# The accuracy of three-dimensional facial scan obtained from three different 3d scanners

**Nichakun Tangthaweesuk**[ID]◉, **Somchart Raocharernporn**[ID]◉*

Department of Oral and Maxillofacial Surgery, Faculty of Dentistry, Mahidol University, Bangkok, Thailand

◉ These authors contributed equally to this work.
* somchart.gj@gmail.com

## Abstract

This study aimed to compare the accuracy (trueness and precision) and reproducibility of three 3D facial scanning systems: a laser scanner (Planmeca Proface), a dual-structured light scanner (EinScan H2), and a smartphone application (EM3D Scanner). Thirty subjects with skeletal deformities scheduled for orthognathic surgery were scanned using these systems, and the resulting 90 3D facial scans were compared with facial surfaces segmented from CBCT scans. Surface discrepancies were measured using root mean square (RMS) values across five facial aesthetic areas (cheeks, nasal, perioral, and mental units) through Geomagic Control X software. The EM3D Scanner showed significantly better trueness and precision compared to the EinScan H2, particularly for the overall face ($p < 0.01$). Planmeca Proface showed no significant difference from the other scanners in terms of error. The nasal and perioral regions, scanned with Planmeca Proface, achieved the highest accuracy compared to other areas, while the left cheek demonstrated the lowest accuracy. Up to 80% of the scanned areas were classified as reproducible, falling within acceptable tolerance limits. Overall, trueness values ranged from 0.70 to 0.85 mm, and precision ranged from 0.68 to 0.81 mm, with deviations of less than 1.0 mm deemed highly acceptable for clinical applications. Surface regions closer to the midline were found to have higher accuracy than those on the sides of the face. These findings highlight the potential of EM3D Scanner and Planmeca Proface for accurate and reliable facial scanning, particularly in clinical settings where minimal deviation is crucial.

## Introduction

Comprehensive diagnosis and treatment planning are essential components of oral and face rehabilitation [1,2]. In craniofacial-maxillofacial surgery, facial morphology study plays an important role for preoperative diagnosis, postoperative evaluation, symmetry analysis, and other purposes. Additionally, it can provide helpful reference values for prosthodontics and orthodontics [3]. Undoubtedly, technological

**Data availability statement:** All relevant data are within the manuscript and its Supporting Information files.

**Funding:** The author(s) received no specific funding for this work.

**Competing interests:** The authors have declared that no competing interests exist.

advancements in Three-dimensional (3D) face scanning devices have revolutionized methods of facial soft tissue analysis. The conventional methods such as direct anthropometry and Two-dimensional photogrammetry (2D) are now being challenged by the superior capabilities of 3D face scanning [4,5]. 3D imaging techniques have recently been increasingly utilized in the medical industry, especially in the fields of maxillofacial surgery and orthodontics. Applications for 3D analysis include the morphology of hard and soft tissues, the research of dentoskeletal relationships, and general facial aesthetics [6,7]. First and foremost, the innovative 3D technology can accurately depict the realistic morphology of both the head and face, enabling simulation, planning, documentation, and prediction of outcomes for patients undergoing orthognathic surgery or maxillofacial reconstruction. Numerous previous studies have demonstrated that the integration of 3D facial photographs with Cone Beam Computed Tomography (CBCT) results in improved accuracy in soft tissue prediction compared to conventional 2D facial prediction methods or using only soft-tissue data provided by CBCT simulations [8–10]. Additionally, 3D faces scanners have the potential to enhance surgical productivity. By utilizing a non-contact measuring instrument, they can significantly reduce time consumption, minimize patient compliance during acquisition, prevent distortion of soft tissue resulting from pressure-related surface changes, and decrease radiation exposure for the patient [11]. For these reasons, 3D facial scanning can accurately predict post-operative facial soft tissue changes. This capability benefits surgeons by aiding in doctor-patient communication and facilitating the determination of treatment options [12]. According to an increasing amount of the literature reports, 3D facial scanners exhibit a high degree of accuracy and precision, making them applicable in the field of dentistry [13–15].

A variety of facial scanning techniques have been developed in recent years, primarily based on operating principles and 3D sensing techniques. These include 3D laser scanning, stereophotogrammetry and structured light scanning [16]. The 3D laser scanner is utilized to capture facial soft tissue by directing a laser beam vertically along the face and detecting the reflected light with a sensor. The image is then reconstructed into three-dimensional data by converting the reflected light into distance information [17]. In dentistry, stationary devices utilizing stereophotogrammetry represent the most prevalent facial scanning technologies. Their reliably reported geometric precision and trueness contribute to their widespread use and acceptance within the field [18–20]. Stereophotogrammetry employs a multi-camera setup to capture two or more images of the same patient simultaneously from different viewpoints. Undoubtedly, this method offers benefits such as minimizing the impact of unintentional head or face movements or expressions on the accuracy of the scan [13]. However, due to limitations and cost considerations, handheld scanning devices employing laser or structured light technologies have emerged as alternatives [21,22]. While numerous professional handheld scanners are regarded as satisfactory in terms of scan image quality, they frequently entail high costs and require substantial training time to become proficient in their complex scanning protocols [3,23,24]. On the other hand, the most recent models of mobile devices, including smartphones and tablets, have cameras built around structured light, a potential 3D

surface imaging reconstruction technology [25–28]. Using the time-of-flight principle—which measures how long it takes light to travel from a sensor array to an object and back again—infrared structured light depth-sensing cameras create a three-dimensional image by rearranging the depth map of the item and its surrounding region [29,30]. Additionally, developers have the capability to create and customize 3D scanning programs using open-source scripts and software coding. Tablet and smartphone devices offer a simple and user-friendly interface for such applications [25,31].

Despite the extensive array of options available on the market and existing literature on the accuracy of specific 3D imaging systems, it remains essential to confirm inter-device accuracy to reliably utilize facial scanning as a clinical tool for diagnostic evaluation. This is achieved by comparing the relative performance of various devices and assessing the scans obtained with them. However, there is still limited research comparing their accuracy in orthodontic patients requiring orthognathic surgery.

Hence, this study aims to analyze and compare the overall and regional accuracy (trueness and precision) ranges between skin surface images derived from CBCT and 3D facial scans, suitable for face scanning in patients with skeletal deformities. Three different types of facial scanning technologies which are particularly promising viable, cost-effective option for general practitioners (GPs): - the Planmeca Promax 3D Proface (Planmeca USA, Inc.; Roselle IL, USA), EinScan H2 (SHINING 3D Tech Co., Ltd., Hangzhou, China), and the EM3D Scanner application version 1.4.1 (Brawny Lads Software, LLC., USA) are included for evaluation and comparison. There were two null hypotheses developed. The first null hypothesis stated that, when scanning the entire face, there is no statistically significant difference in accuracy between scanners. The second null hypothesis posited that there exists no statistically significant in accuracy between scanners when examining distinct facial regions.

## Materials and methods

### Study sample

This study was approved by The Ethics Committee of Mahidol University (COA.No.MU-DT/PY-IRB 2021/005.2701.) The World Medical Association's Helsinki Declaration and the STROBE guidelines were followed throughout the study. All patients included via consecutive sampling method provided written informed consent for study participation. The individuals in this manuscript provided written informed consent to publish these case details. The study group consisted of 30 patients (18–43 years old, mean±SD: 28.03±6.84 years), 18 females and 12 males, with Skeleton deformity and scheduled for orthognathic surgery in the Oral and Maxillofacial surgery clinic of Mahidol University (Bangkok, Thailand). Class III Skeleton deformity patients (n = 21), Class II Skeleton deformity patients (n = 5), Class I skeleton asymmetry (n = 3) and Class I skeleton deformity with bimaxillary protrusion (n = 1) are selected for this study from February 2022 to December 2022. The surgery provided for the patient was Bilateral sagittal Split Ramus osteotomy with or without Lefort I for the correction of Skeleton deformity. Patients with a cleft, craniofacial deformity, jaw defect, face muscle spasm symptoms and unstable tooth contacts in intercuspal occlusion (ICP) were excluded.

### Experimental instrument

A variety of facial scanners are used in scientific and commercial settings. In this study, the facial scanners allocated for assessment included the Planmeca Promax 3D Proface (Planmeca USA, Inc.; Roselle IL, USA), EinScan H2 (SHINING 3D Tech Co., Ltd., Hangzhou, China), and the EM3D Scanner application version 1.4.1 (Brawny Lads Software, LLC., USA)

The Planmeca ProMax 3D ProFace is a CBCT imaging unit that incorporates an integrated 3D face scan system. This imaging unit, enhanced with ProFace technology, can capture both a 3D photo and a CBCT image in a single rotation. Alternatively, the 3D photo can be obtained independently. This process is entirely radiation-free, as lasers scan the facial geometry, while digital cameras capture the color texture of the face.

The EinScan H2 is a handheld 3D scanner utilizing a Hybrid LED and Infrared Light Source. This dual-light approach enhances scanning efficiency, with the LED light providing swift 3D scanning and precise, high-quality data. The Infrared VCSEL is well-suited for capturing dark surfaces, human body scanning, and environments with bright lighting. The scan accuracy is within the range of 0.05mm to 0.1mm. Additionally, it can generate 3D face models in OBJ, STL, ASC, PLY, P3, and 3MF formats with authentic color texture.

The EM3D Scanner application utilizes the TrueDepth camera integrated into Apple devices (Apple Store, Cupertino, CA, USA) running on iOS 13.0 and later, specifically designed for iPad Pro or iPhone models equivalent to or surpassing iPhone X for conducting scans. Using this device is similar to using a handheld 3D scanner, allowing the application to rapidly generate a 3D representation of a subject's face in less than 15 seconds, capturing measurements from diverse angles. Furthermore, it possesses the capacity to generate 3D face models in OBJ, STL, and PLY formats, inclusive of genuine color texture. In our study, we utilized an iPhone 13 (Apple Store, Cupertino, CA, USA) with the EM3D Scanner application version 1.4.1 (Brawny Lads Software, LLC., USA) to collect data for this research.

## Face model acquisition

Morphologic points necessary for localized surface areas were identified through visual inspection and palpation, followed by marking on the face. All landmarks were marked using marker stickers designed for the 3D scanner. These stickers feature black rings and centers on a white background, facilitating easy identification of the black centers in 3D scanning images. The marking of morphologic points was consistently performed by a qualified examiner with expertise and certification in the field of oral and maxillofacial surgery. A total of nine markers were strategically placed on the patient's face in specified positions: (Table 1)

This study was performed with 90 3D photographs and 30 CBCT scans for the same subjects. The Planmeca ProMax® 3D Mid Cone Beam scanner (Planmeca USA, Inc.; Roselle IL, USA) was utilized for the scanning procedure. Subjects' heads were positioned in a natural head posture with the face directed forward, and their teeth were in centric occlusion. Temple supports were employed to stabilize the patients' heads, and no additional instruments such as chin caps, headbands, or biting rods were used to prevent soft tissue distortion and interference with the occlusal plane. The CBCT data for each individual patient were exported in Digital Imaging and Communications in Medicine (DICOM) format on a CD-ROM.

Subsequently, three-dimensional virtual models were generated from the CBCT data through segmentation using Simplant imaging-3D (SimPlant Ortho Pro software, version 2.0, Materialise Dental, Leuven, Belgium).

Each patient underwent three face-scanning operations, with data from three different scanners collected on the same day. All 3D face scanners were used to collect data within the same closed environment, specifically in the imaging room for cone beam computed tomography (CBCT) located in the Department of Radiology, Faculty of Dentistry, Mahidol

**Table 1. Description of 3D soft tissue landmarks in the study.**

| Landmark | Definition |
|---|---|
| Pronasale (Pn) | The most protruded point of the apex nasi. |
| Orbitale (Or' Rt/ Lt) | The lowest point on the inferior margin of the orbit on right side/left side. |
| Pogonion' (Pog') | The most anterior midpoint of chin |
| Menton' (Me') | The lowest median landmark on the lower border of the mandible |
| Gonion (Go' Rt/ Lt) | The midpoint between the most posterior and inferior points of the mandibular angle on right side/left side. |
| Cheek point (Ck Rt/Lt) | The point where a vertical line from exocanthion and a horizontal line from cheilion meet on right side/left side. |

University. This room was a fully enclosed space with no windows, ensuring that no natural light enters the environment. The room was illuminated solely by LED 22W Daylight bulbs. To ensure that each patient maintained the same expression for the three scanning processes, the following measures were adopted: every patient was strictly supervised by the well-trained examiners throughout the entire scanning process to maintain a sitting position, stable support for head and neck, natural head position (NHP), intercuspal position (ICP, a stable mandibular position) eyes and lips closing naturally and relaxed body. Patients were instructed to remove glasses, earrings, and necklaces for reducing a source of image artifacts. Any shiny surfaces, primarily due to oily skin, or cosmetics were removed and a light dusting of powder was applied around the nose, ear and forehead can reduce shininess.

The first scan was conducted using the Planmeca ProMax 3D ProFace (Planmeca USA, Inc.; Roselle IL, USA) which has the capability to capture both a 3D photo and a CBCT image simultaneously. Following that, the patient's facial model underwent 3D scanning using two additional scanners in accordance with the manufacturer's recommendations: EinScan H2 (SHINING 3D Tech Co., Ltd., Hangzhou, China), and iPhone 13 (Apple Store, Cupertino, CA, USA) using the EM3D Scanner application version 1.4.1 (Brawny Lads Software, LLC., USA).

To use the EinScan H2 scanner to scan a subject's face, the method involves connecting the handheld EinScan H2 device to the software, particularly the one provided with the device, through a laptop. It is important to ensure an unobstructed view of the display that shows the area currently being captured by the scanner. This allows for proper monitoring and control of the scanning process. Scanning was conducted as swiftly as possible by our trained operator, ensuring there were no delays in the procedure. The scanning direction started from the right side of the subject's face and rotated clockwise until obtaining the desired images of the subject's face. To use the EM3D Scanner application, the subject holds the iPhone with their right hand, treating it like an arm device, to ensure the distance is as consistent as possible on all sides and to reduce shaking while moving the device. Then, the same operator guides the subject's arm holding the device to determine the scanning direction of the face. The scanning starts from the right side of the face and rotates clockwise until obtaining the desired images of the subject's face on the iPhone screen.

To prevent results from being distorted by any long-term changes in the patient's face, both types of scans were taken at approximately the same time. All facial scans were conducted by a single operator with experience in handling scanning devices. The scan data files were saved in stereolithography (STL) format for subsequent measurements.

## Modification, trim and division of face models

Our research's second phase consisted of modifying and setting the file scans in the ideal locations so that they overlapped and could be used to analyze facial points, areas, and characteristics in the future. There were multiple steps in this process shown in Fig 1.

This study compared 90 facial 3D scans with 30 corresponding CBCT-segmented facial surfaces. The three face models of each patient were trimmed in the same boundary using Meshmixer software (MeshmixerTM Autodesk®, Inc., San Rafael, CA, USA) to further reduce the influence on 3D accuracy analysis of the noise data on face models (such as hair and ear data). The appearance of hair is typically unstable, and the area around the ears is particularly susceptible to the influence of hair. Large scanning errors consistently occur in these areas, which cannot be attributed to the scanner's capabilities. Furthermore, we eliminated these locations from our research to cleanse the data and saved overlapping parts in each group for accuracy analysis, considering the minimal attention that has been given to these areas.

We imported the facial 3D scans of the patients and the above-modified final CBCT scans in the Geomagic X Control software (3D Systems Inc, Rock Hill, SC, USA). For the purpose of evaluating the specimens' accuracy based on the total or localized surface areas of the CBCT segmentation. Using this specific program, the control mesh was marked with different the facial aesthetic units which were based on Gonzales-Ulloa's original work that separated the CBCT segmentation into 5 different areas (from Area-1 to Area-5) as shown in Fig 2 and Table 2.

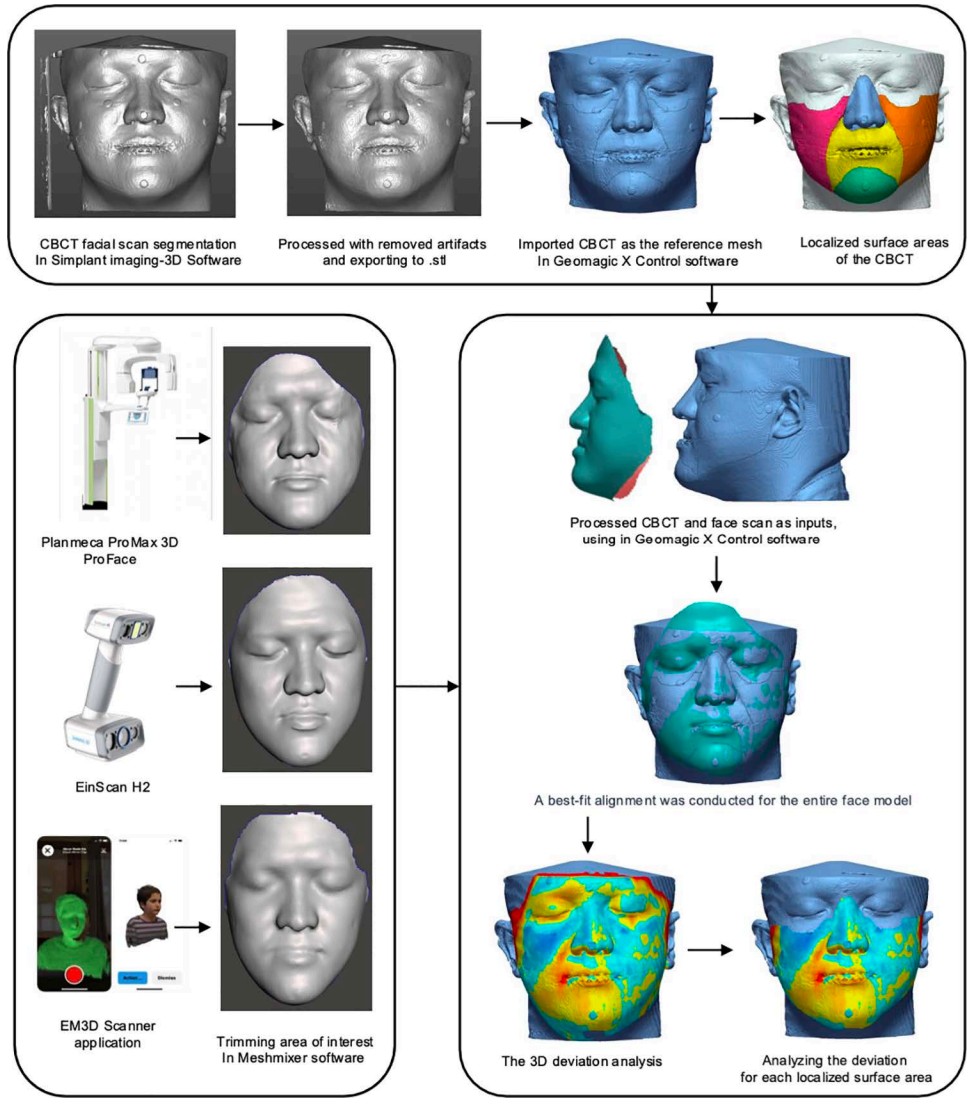

**Fig 1. The diagrammatic representation of the proposed method.**

We decided to limit the usage of the facial aesthetic unit only in the midface and lower face areas, as these are parts of the face that undergo changes after orthognathic surgery.

## Data analysis and comparison

The test models from 3D face scan (Planmeca ProFace, EinScan H2, and EM3D Scanner application) and the reference (CBCT segmentation) were compared in three dimensions using the "3D compare" feature in the Geomagic X Control software. This software facilitated the superimposition of the reference and each test model through alignment processes. Specifically, the alignment was achieved using the best-fit algorithm, which automatically determined the optimal alignment by utilizing pre-established reference points. To investigate the qualitative congruency of the reference and test models, color difference images were produced. The root mean square (RMS) of all distances between the closest point pairs

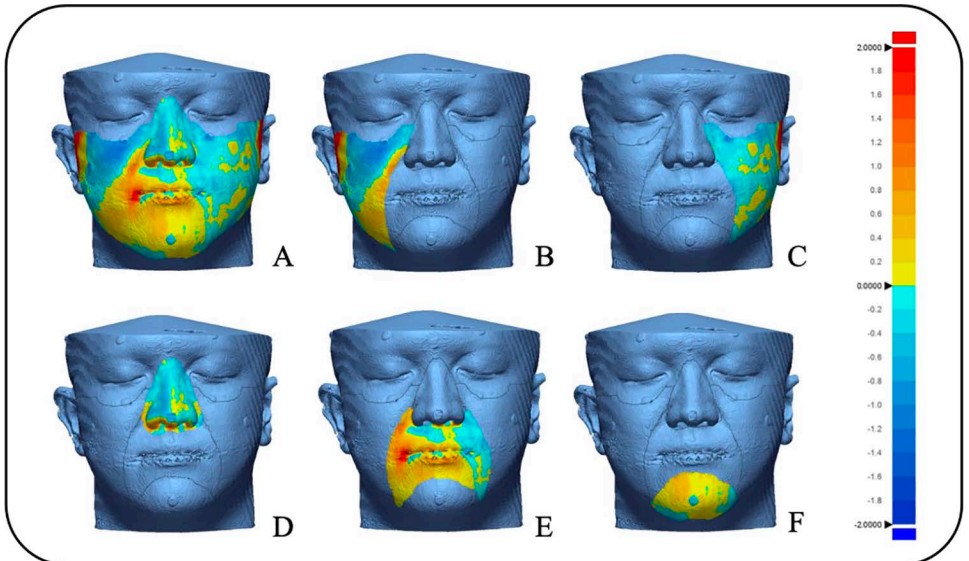

**Fig 2. Soft tissue boundaries were outlined on the reference mesh to assist in posterior measurements.** A, Total face surface. B, Frontal view of the areas from 1 to 5. C, Lateral view of the areas from 1 to 5.

on the reference and test models was computed to determine the "3D deviation" between each pair for the total surface facial scan area and for each specific area (from Area-1 to Area-5), as shown in Fig 3.

The software's algorithm automatically found and matched the nearest point pairs. In this investigation, good test model 3D accuracy was indicated by low RMS scores, which show a strong 3D congruency of the superimposed models.

Intersystem evaluation: Trueness was defined as the mean of the average absolute dimensional discrepancy between the reference mesh and the facial scans (RMS). In other words, trueness evaluates how closely the digitized object aligns with its true dimensions.

Intrasystem evaluation: Precision was described as the standard deviation (SD) of the dimensional discrepancies between the reference mesh and the facial scans. Therefore, precision analyzes the reproducibility of the facial scanner [32].

The reliability of a digital face scanner can be classified into 4 categories [33]:

**Table 2. Description of surface deviations areas for aesthetic units.**

| Surface deviations area | Soft tissue landmark and boundaries |
|---|---|
| Area-1 (Right Cheek unit) | Surface located between the superiorly by the infraorbital rims (Or' Rt) and the superior aspects of the zygomatic arches, laterally by the preauricular creases, inferiorly by the jaw line, and medially by the nasolabial and melolabial (labiomandibular) grooves and the lateral aspects of nasal dorsal side walls. |
| Area-2 (Left Cheek unit) | Surface located between the superiorly by the infraorbital rims (Or' Lt) and the superior aspects of the zygomatic arches, laterally by the preauricular creases, inferiorly by the jaw line, and medially by the nasolabial and melolabial (labiomandibular) grooves and the lateral aspects of nasal dorsal side walls. |
| Area-3 (Nasal unit) | Surface located between the nasion superiorly, junction of the cheeks and nasal dorsal side walls laterally, and the alar groove and columella inferiorly. |
| Area-4 (Perioral unit) | Surface located between the alar grooves and columella superiorly, the nasolabial grooves laterally, the melolabial grooves laterally, and the mentolabial groove inferiorly. |
| Area-5 (Mental unit) | Surface located between the mentolabial groove superiorly, forming a curvilinear border laterally, and ending at the submental crease, just inferoposterior to the jaw line. |

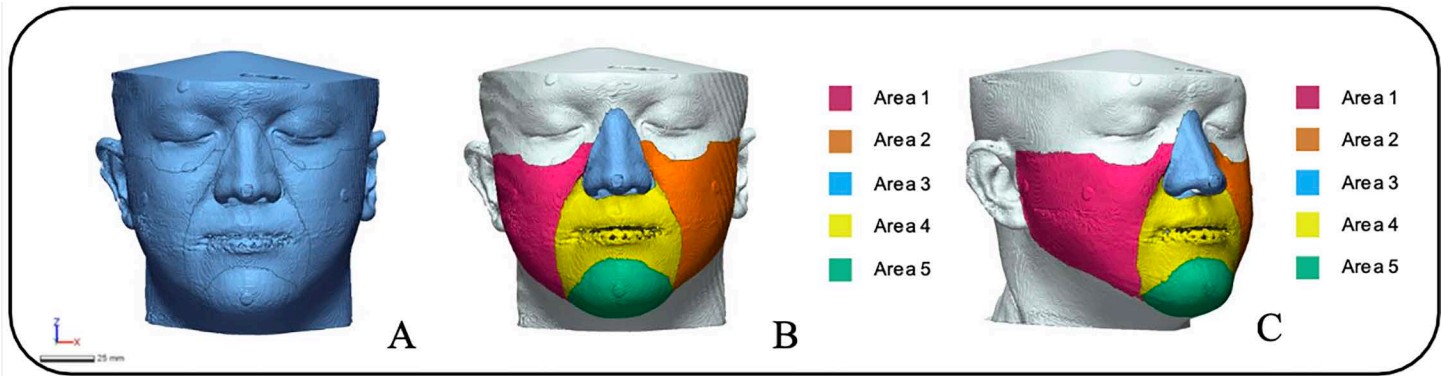

**Fig 3. Representative color-coded deviation map showing the discrepancy between the reference mesh and a facial scan using the best-fit algorithm.** A, Overall surface area. B, From area -1. C, From area -2. D, From area -3. E, From area -4. F, From area -5.

- Highly reliable (deviation <1.0 mm)

- Reliable (deviation 1.0 mm-1.5 mm)

- Moderately reliable (deviation 1.5 mm-2.0 mm)

- Unreliable (deviation >2.0 mm)

Furthermore, the data are presented as the average percentage of overlapping surfaces between each 3D facial scanner and the reference model, analyzed across five distinct areas as defined by the software. The tolerance ranges, which represent the deviations between the test models and the reference model, are divided into five predefined intervals. These intervals indicate the areas where each 3D facial scanner achieves high reproducibility and alignment accuracy. The details are outlined as follows:

- -0.5mm to 0mm and 0mm to 0.5mm (highly reproducible)

- -1 mm to -0.5 mm and 0.5 mm to 1 mm (moderately reproducible)

- -1.5 mm to 1- mm and 1 mm to 1.5 mm (poorly reproducible)

- >1.5 mm (not reproducible)

- <-1.5 mm (not reproducible)

For intraexaminer reliability assessment, ten CBCT soft tissue segmentations were randomly selected. Each segmentation was manually marked for five localized surface areas twice by the same examiner, with a minimum interval of seven days between each marking process. Additionally, the comparison reports were analyzed twice, also with an interval of seven days. The values were then analyzed using the Intraclass Correlation Coefficient (ICC).

"The individual pictured in Figs 1–3 has provided written informed consent (as outlined in PLOS consent form) to publish their image alongside the manuscript".

## Statistical analysis

Data were analyzed using IBM SPSS Statistics, Version 29.0 (Armonk, NY: IBM Corp). The intraclass correlation coefficient (ICC) was employed to assess the consistency and repeatability of the localized surface area of the reference mesh method. An ICC value greater than 0.9 was considered indicative of excellent reliability.

To examine the data distribution for trueness and precision, the Kolmogorov-Smirnov and Shapiro-Wilk normality tests were applied to all datasets, which included three groups, each consisting of ten computed values. When necessary, the data for trueness and precision were transformed to achieve normal distribution.

To evaluate the accuracy of each scanner across the entire face and five specific areas, the means and standard deviations (SD) for trueness (RMS) and precision (SD) were calculated. Differences in trueness and precision across the three groups were analyzed using one-way repeated measures ANOVA. Post-hoc comparisons were performed using the Bonferroni test, a statistical procedure for comparing multiple pairs of means within the data. A p-value of $< 0.05$ was considered statistically significant.

Finally, tolerance values were analyzed as a function of the five ranges and the face side using descriptive analysis. The resulting model provided average percentages and estimated confidence intervals (lower and upper confidence limits, CL) for the tolerance measures of overlapping surfaces. These were calculated for comparisons between each 3D facial scanner and CBCT soft tissue segmentation. For all evaluations, a p-value of $< 0.05$ was considered statistically significant.

## Results

The intraexaminer reliability for the five localized surface areas (as shown in Table 3.) was high, with all yielding an ICC index greater than 0.9. This suggests substantial consistency in the markings made by the examiner.

The overall mean values for trueness and precision associated with each 3D facial scanner are presented in Table 4. Additionally, Tables 5 and 6 provide the mean values for trueness and precision for each facial scanner divided by region.

### Overall accuracy (trueness and precision)

For trueness, the Planmeca ProFace group obtained an overall mean±SD RMS value of 0.75±0.18 mm. In comparison, the Einscan H2 obtained an overall trueness value of 0.85±0.28 mm, and the EM3D Scanner application recorded an overall trueness value of 0.70±0.16 mm.

The mean trueness values of the Planmeca ProFace, EinScan H2, and EM3D Scanner applications revealed significant differences, as indicated by the One-way repeated measures ANOVA analysis and Bonferroni's pairwise comparison. Pairwise comparisons indicated that the EM3D Scanner application exhibited significantly smaller errors compared to the

**Table 3. The Intraclass Correlation Coefficient (ICC) index and F-test of averages were calculated for the five localized surface areas as marked.**

| Surface area | ICC[a] (95% CI[b]) | F test with true value | |
|---|---|---|---|
| | | Value | p-Value |
| Area-1 (Right Cheek unit) | 0.99 (0.97–1) | 130.38 | <.001[*] |
| Area-2 (Left Cheek unit) | 1 (0.99–1) | 293.87 | <.001[*] |
| Area-3 (Nasal unit) | 0.98 (0.91–0.99) | 45.28 | <.001[*] |
| Area-4 (Perioral unit) | 0.99 (0.96–1) | 103.59 | <.001[*] |
| Area-5 (Mental unit) | 0.94 (0.77–0.99) | 17.82 | <.001[*] |

[*]p<0.05

a Intraclass correlation coefficient.

b 95% confidence interval

EinScan H2 scanners (p<0.01). Following this, the Planmeca ProFace showed no significant difference in error compared to the other scanners. Finally, the EinScan H2 demonstrated statistically significantly lower trueness (p<0.01) compared to the EM3D Scanner application concerning the overall face (Table 4, Fig 4).

In terms of precision, the Planmeca ProFace group obtained a mean±SD RMS value of 0.72±0.17 mm overall. By contrast, the EM3D Scanner application recorded an overall precision value of 0.68±0.16 mm, while the Einscan H2 obtained an overall precision value of 0.81±0.26 mm. The One-way repeated measures ANOVA analysis and Bonferroni's pairwise comparison showed significant difference between mean Precision values from the Planmeca ProFace, EinScan H2, and EM3D Scanner application. Pairwise comparisons unveiled that the EM3D Scanner application displayed a statistically significantly higher precision compared to the EinScan H2 devices (p<0.01). Furthermore, the EinScan H2 exhibited less precision than the other tested devices, although there was no statistically significant difference with the Planmeca ProFace. (Table 4, Fig 5).

**Table 4. Descriptive statistics of the overall scanning accuracy (trueness and precision) obtained using three different 3D scanners (Planmeca ProFace, EinScan H2, and EM3D Scanner application). Data provided in millimetres (mm).**

| Experimental instrument | Trueness | | F-test | P-value | Partial Eta Squared | Precision | | F-test | P-value | Partial Eta Squared |
|---|---|---|---|---|---|---|---|---|---|---|
| | Mean | SD | | | Mean | SD | | | | |
| Planmeca ProFace | 0.75[ab] | 0.18 | 5.051 | 0.015* | 0.305 | 0.72[ab] | 0.17 | 4.622 | 0.021* | 0.287 |
| EinScan H2 | 0.85[a] | 0.28 | | | | 0.81[a] | 0.26 | | | |
| EM3D Scanner application | 0.70[b] | 0.16 | | | | 0.68[b] | 0.16 | | | |

*p<0.05

SD, Standard deviation.

a, b post-hoc comparisons were performed using the Bonferroni test. when the superscripts (a,b) are the same, it indicates no statistically significant difference between the compared pairs of data.

**Table 5. Descriptive statistics of the scanning accuracy (trueness) values (RMS) obtained among the different areas. Data provided in millimeters (mm).**

| Surface area | Planmeca ProFace | | EinScan H2 | | EM3D Scanner application | | F-test | P-value | Partial Eta Squared |
|---|---|---|---|---|---|---|---|---|---|
| | Mean | SD | Mean | SD | Mean | SD | | | |
| Area-1 (Right Cheek unit) | 0.77 | 0.24 | 0.85 | 0.30 | 0.68 | 0.21 | 2.433 | 0.112 | 0.188 |
| Area-2 (Left Cheek unit) | 0.92[a] | 0.33 | 0.87[ab] | 0.34 | 0.73[b] | 0.22 | 3.603 | 0.042* | 0.217 |
| Area-3 (Nasal unit) | 0.36[b] | 0.08 | 0.52[a] | 0.19 | 0.57[a] | 0.18 | 9.939 | 0.002* | 0.554 |
| Area-4 (Perioral unit) | 0.50[b] | 0.17 | 0.69[a] | 0.25 | 0.69[a] | 0.20 | 6.988 | 0.004* | 0.359 |
| Area-5 (Mental unit) | 0.54 | 0.17 | 0.60 | 0.22 | 0.66 | 0.19 | 2.276 | 0.130 | 0.193 |

*p<0.05

SD, Standard deviation.

a, b post-hoc comparisons were performed using the Bonferroni test. when the superscripts (a,b) are the same, it indicates no statistically significant difference between the compared pairs of data.

**Table 6. Descriptive statistics of the scanning accuracy (precision) values (RMS) obtained among the different areas. Data provided in millimeters (mm).**

| Surface area | Planmeca ProFace | | EinScan H2 | | EM3D Scanner application | | F-test | P-value | Partial Eta Squared |
|---|---|---|---|---|---|---|---|---|---|
| | Mean | SD | Mean | SD | Mean | SD | | | |
| Area-1 (Right Cheek unit) | 0.70 | 0.22 | 0.68 | 0.22 | 0.62 | 0.18 | 1.034 | 0.373 | 0.090 |
| Area-2 (Left Cheek unit) | 0.84[a] | 0.30 | 0.68[ab] | 0.25 | 0.66[b] | 0.19 | 3.557 | 0.043[*] | 0.215 |
| Area-3 (Nasal unit) | 0.30[b] | 0.05 | 0.50[a] | 0.19 | 0.52[a] | 0.18 | 14.958 | <0.001[*] | 0.652 |
| Area-4 (Perioral unit) | 0.44[b] | 0.15 | 0.61[a] | 0.20 | 0.65[a] | 0.18 | 9.484 | 0.001[*] | 0.431 |
| Area-5 (Mental unit) | 0.46[b] | 0.14 | 0.50[ab] | 0.20 | 0.59[a] | 0.18 | 4.202 | 0.031[*] | 0.307 |

[*]p < 0.05

SD, Standard deviation.

a, b post-hoc comparisons were performed using the Bonferroni test. when the superscripts (a,b) are the same, it indicates no statistically significant difference between the compared pairs of data.

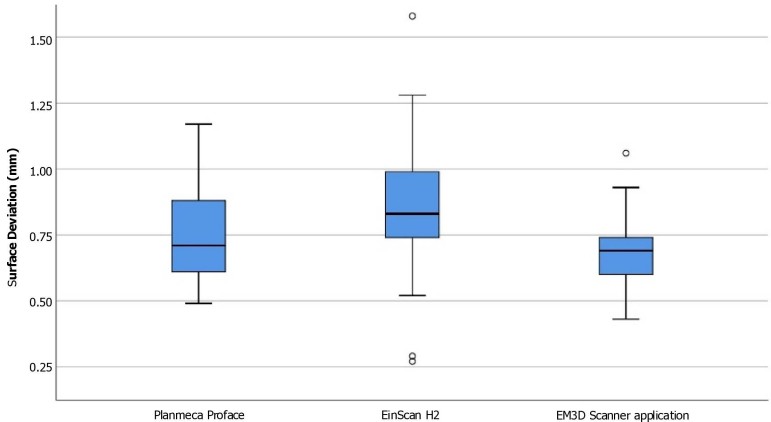

**Fig 4. Boxplot of the overall scanning accuracy (trueness) obtained using three different 3D scanners.**

## The accuracy (trueness and precision) varied among the different areas

All the localized surface areas examined in this study exhibited a significant difference in the trueness values across surface areas. When comparing the regions, the Planmeca ProFace exhibited the highest mean trueness value, which was 0.92 mm in the Left Cheek unit (Area-2). The nasal unit (Area-3) exhibited the lowest mean trueness (mean = 0.36 mm) calculated by the Planmeca ProFace, suggesting that the localized surface of the nasal unit achieved the highest trueness score. Additionally, statistically significant differences were found between the various measurements of all three facial scanners, except for the right cheek unit (Area-1) and mental unit (Area-5), according to One-way repeated measures ANOVA and Bonferroni's pairwise tests (Table 5, Fig 6).

Additionally, when comparing the different localized surface areas captured by the three face scanners, it was observed that the precision (SD) remained under 1.00 mm. Notably, in the Left cheek unit (Area-2), the Planmeca ProFace exhibited the highest precision value (mean = 0.84 mm). Conversely, for the nasal unit (Area-3), the Planmeca ProFace produced

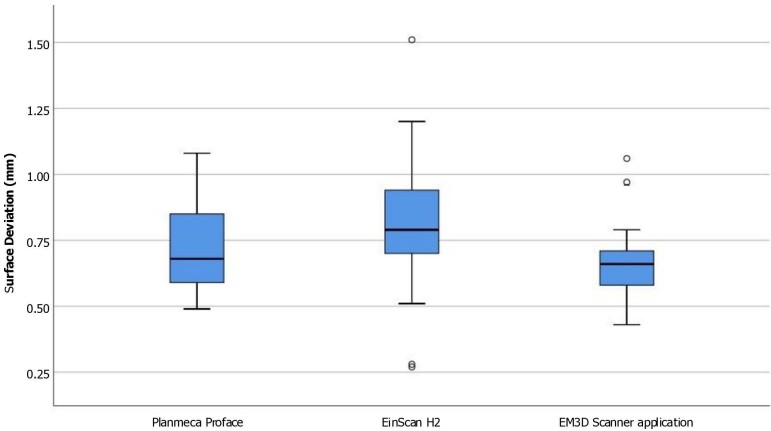

**Fig 5. Boxplot of the overall scanning accuracy (precision) obtained using three different 3D scanners.**

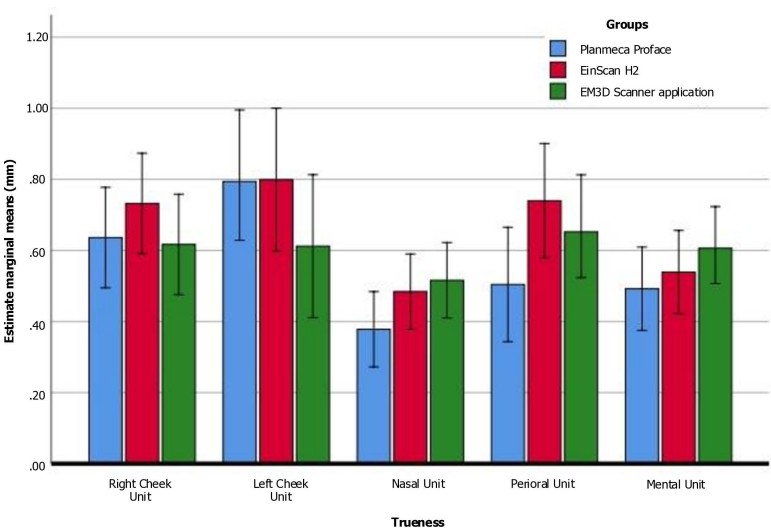

**Fig 6. Mean of the scanning accuracy (trueness) values (RMS) obtained among the different areas.**

the lowest precision values (mean = 0.30 mm), indicating that the localized surface of the nasal unit area achieved a higher precision score. Furthermore, One-way repeated measures ANOVA and Bonferroni's pairwise testing revealed statistically significant differences between the various measurements of all three facial scanners, except for the right cheek unit (Area-1) (Table 6, Fig 7).

The average percentage overlap of the 3D-scanned surfaces across the five bands of tolerance for the five distinct areas, along with the corresponding confidence intervals (lower CL and upper CL) between each 3D facial scanner and CBCT soft tissue segmentation, were presented in Table 7.

According, to Table 7. For each of the three comparisons that were examined, between 80 and 90 percent of the overlapping areas were inside the tolerance limits. The nasal unit (Area-3) is the area with the fewest values outside the

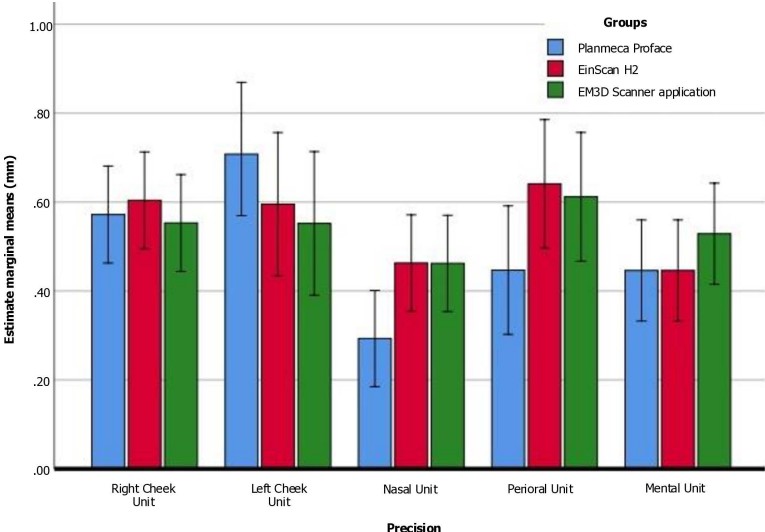

**Fig 7. Mean of the scanning accuracy (precision) values (RMS) obtained among the different areas.**

tolerance limits (> 1.5 mm and < − 1.5 mm). Even though there were more participants with a percentage of overlap in the non-reproducible bands for the Surface area, these values never involved more than 10% of the area—except for one comparison where the Left cheek unit (Area-2) in the Einscan H2 showed more than 10% of overlapping area>1.5 mm.

## Discussion

Face scanners have become a prominent topic of discussion, with numerous authors examining the advancements of this technology in the medical field in recent years. Hence, the predominant choice for evaluating the accuracy of 3D scanners involves using inanimate mannequins as study subjects. This aims to mitigate any image distortions associated with involuntary movements [28,34–36]. However, it is important to acknowledge the limitation of an in vitro study, as patients typically cannot maintain absolute stillness in clinical settings, which may affect the study's applicability. Further-more, previous reviews have similarly highlighted the significance of artifacts caused by living subjects, underscoring their importance in research [37]. Hence, this study specifically concentrated on live subjects with skeleton deformity, aiming to delineate the attributes of various technologies utilized in clinical practice. The aim of this study is to assess three different face scanning systems: laser scanners (Planmeca ProFace), dual-structured light (EinScan H2), and the TrueDepth camera system found on Apple iPhones (EM3D Scanner application). A prior investigation categorized the digital face scanner into four groups: highly reliable (deviation <1.0 mm), reliable (deviation 1.0 mm -1.5 mm), moderately reliable (deviation 1.5–2.0 mm), and unreliable (deviation >2.0 mm). Statistically significant differences in scanning accuracy were observed among the three facial scanning modalities for both the overall face and the five separate regions of the face (Area-1 to Area-5). Consequently, both null hypotheses were rejected. In clinical applications, particularly for tasks like facial aes-thetic analysis in patient diagnosis and treatment planning, deviations less than 1.0 mm were deemed highly acceptable [3,31,38,39]. The superimposition procedures that were tested exhibited notable variances in both trueness and preci-sion RMS values. The tested scanner yielded an overall trueness ranging from 0.70 to 0.85 mm and an overall precision ranging from 0.68 to 0.81 mm. Consequently, the computed accuracy values fall within the clinically acceptable scanning accuracy range.

In this study, the EM3D scanner application exhibited the highest overall accuracy performance in both the middle face and lower face areas, with trueness values of 0.70 ± 0.16 mm and precision values of 0.68 ± 0.16 mm. However, when

**Table 7. The average percentages and estimated confidence intervals of the tolerance measures for overlapping surfaces in five different areas across five tolerance bands, which were compared between each 3D facial scanner and CBCT soft tissue segmentation.**

| Tolerance ranges | Surface area | Mean %[a] (95% CI[b]) | | |
|---|---|---|---|---|
| | | Planmeca ProFace | EinScan H2 | EM3D Scanner application |
| TOL: 0.5 to 0; 0 to -0.5 (mm) | Right cheek unit | 62.16 (57.51-66.82) | 47.55 (39.76-55.36) | 52.94 (46.76-59.12) |
| TOL: 1 to 0.5; -0.5 to -1 (mm) | Right cheek unit | 22.22 (19.10-25.35) | 27.70 (23.66-31.74) | 28.35 (24.23-32.48) |
| TOL: 1.5 to 1; -1 to -1.5 (mm) | Right cheek unit | 7.20 (5.44-8.96) | 14.68 (11.28-18.08) | 10.89 (8.14-13.66) |
| TOL: > 1.5 (mm) | Right cheek unit | 7.75 (5.39-10.19) | 9.55 (6.34-12.76) | 2.53 (0.88-4.20) |
| TOL: < 1.5 (mm) | Right cheek unit | 0.67 (0.24-1.10) | 0.52 (0.01-1.04) | 5.27 (1.30-9.23) |
| TOL: 0.5 to 0; 0 to -0.5 (mm) | Left cheek unit | 60.61 (55.65-60.84) | 46.55 (38.49-54.61) | 54.96 (49.27-60.66) |
| TOL: 1 to 0.5; -0.5 to -1 (mm) | Left cheek unit | 21.17 (18.07-24.26) | 26.95 (22.56-31.35) | 26.39 (23.32-29.48) |
| TOL: 1.5 to 1; -1 to -1.5 (mm) | Left cheek unit | 7.52 (5.95-9.08) | 15.56 (11.40-19.71) | 10.61 (7.99-13.24) |
| TOL: > 1.5 (mm) | Left cheek unit | 9.79 (6.65-12.93) | 10.26 (6.29-14.24) | 3.95 (1.67-6.23) |
| TOL: < 1.5 (mm) | Left cheek unit | 0.91 (0.40-1.43) | 0.67 (0.23-1.33) | 4.07 (1.42-6.73) |
| TOL: 0.5 to 0; 0 to -0.5 (mm) | Nasal unit | 79.12 (73.29-84.95) | 61.97 (53.86-70.08) | 57.83 (50.85-64.80) |
| TOL: 1 to 0.5; -0.5 to -1 (mm) | Nasal unit | 18.34 (13.56-23.13) | 24.96 (21.00-28.93) | 24.46 (21.14-27.78) |
| TOL: 1.5 to 1; -1 to -1.5 (mm) | Nasal unit | 2.18 (1.12-3.23) | 8.62 (5.39-11.85) | 10.23 (7.52-12.93) |
| TOL: > 1.5 (mm) | Nasal unit | 0.33 (0.30-0.63) | 3.25 (0.93-5.57) | 3.42 (1.21-5.65) |
| TOL: < 1.5 (mm) | Nasal unit | 0.039 (-0.01-0.07) | 1.19 (0.39-1.99) | 4.06 (1.46-6.66) |
| TOL: 0.5 to 0; 0 to -0.5 (mm) | Perioral unit | 67.82 (61.97-73.65) | 52.93 (46.22-59.63) | 52.17 (45.74-58.60) |
| TOL: 1 to 0.5; -0.5 to -1 (mm) | Perioral unit | 21.99 (18.48-25.50) | 27.88 (24.91-30.85) | 28.13 (25.28-30.98) |
| TOL: 1.5 to 1; -1 to -1.5 (mm) | Perioral unit | 5.57 (3.64-7.51) | 12.36 (9.26-15.46) | 11.96 (9.67-14.25) |
| TOL: > 1.5 (mm) | Perioral unit | 3.43 (0.87-5.99) | 3.34 (1.36-5.32) | 4.44 (1.47-7.40) |
| TOL: < 1.5 (mm) | Perioral unit | 1.19 (0.65-2.31) | 3.49 (1.23-5.76) | 3.30 (1.66-4.94) |
| TOL: 0.5 to 0; 0 to -0.5 (mm) | Mental unit | 68.41 (60.75-76.06) | 55.32 (47.29-63.35) | 52.66 (45.11-60.21) |
| TOL: 1 to 0.5; -0.5 to -1 (mm) | Mental unit | 22.50 (17.01-27.99) | 28.92 (24.81-33.03) | 25.74 (23.02-28.46) |
| TOL: 1.5 to 1; -1 to -1.5 (mm) | Mental unit | 5.91 (3.26-8.56) | 10.69 (7.27-14.11) | 12.77 (9.53-16.01) |
| TOL: > 1.5 (mm) | Mental unit | 1.97(-0.13-4.07) | 2.01 (0.14-3.88) | 5.74 (2.67-8.80) |
| TOL: < 1.5 (mm) | Mental unit | 1.22 (0.52-1.91) | 3.05 (0.21-5.90) | 3.09 (1.13-5.06) |

a Obtained from the marginal means of the descriptive analysis.

b 95% confidence interval, TOL indicates tolerance.

investigating the scanning trueness and precision of the tested devices across five different localized surface areas of the face, it was found that the EM3D scanner application exhibited greater trueness value compared to both the EinScan H2 and Planmeca ProFace scanners, particularly in the cheek unit area. Furthermore, statistical analysis showed that there was no significant difference in trueness between the EM3D scanner application and the other scanners, particularly in the mental unit area. As of now, there are no existing studies in the literature investigating the scanning accuracy of the EM3D scanner application. This is attributed to its recent introduction to the 3D face scanner market in 2020. According to Amornvit et al. (2019) [28], a smartphone's 3D depth-sensing sensor scanner is accurate when measuring linearly in the frontal plane, but it is less accurate when measuring depth when compared to professional face scanners. These findings align with those of Liu et al. (2019) [40], who concluded that professional 3D facial scanners and mobile device-compatible face scanners perform similarly for simple and flat facial regions such as the forehead, cheeks, and chin. However, when complex facial regions such as the external ears, eyelids, nostrils, and teeth were scanned using mobile device-compatible face scanners, the scanning accuracy was notably low [28,41,42]. Depending on the depth of the fault, facial regions with defects exhibited higher levels of inaccuracy [25]. The results from our study indicate that

there are differences in accuracy between the right and left cheek units. Specifically, the right cheek unit shows no statistically significant differences between the various measurements of all three facial scanners. However, the left cheek unit demonstrates statistically significant differences between the EM3D scanner application and the Planmeca Proface scanner. In this case, the researchers speculate that the differences in accuracy between the right and left cheek units may stem from the data collection method used by the EM3D scanner application. When subjects use their right arm to hold the device, it may not capture the left side of the face comprehensively during movements, particularly the left cheek area. This incomplete coverage could result in certain parts of the facial surface data being replaced by the program during the generation of the 3D facial reconstruction. Consequently, the left side of the face may not accurately reflect reality, affecting accuracy when compare in the research study. Therefore, when using face scanners compatible with mobile devices, careful consideration based on the purpose and the individual's characteristics may be necessary. The mean discrepancy values of scanned faces obtained utilizing mobile device-compatible 3D facial scanners ranged from 0.34 to 1.40 mm in the studies that Mai HN & Lee DH (2020) [43] analyzed based on their systematic reviews and meta-analysis. Furthermore, the meta-analysis revealed that professional 3D facial scanners were shown to be more accurate than 3D scanners compatible with mobile devices. Our findings are consistent with those of Kühlman et al. (2023) [44], as they also observed that when comparing all four scan applications based on 3D depth-sensing cameras, the mean absolute differences and standard deviations among the ten different areas of the face were all less than 1.0 mm. Additionally, the scan accuracy (both trueness and precision) of depth measurements for all scan applications showed values less than 1.00 mm. In the context of clinical applications, a deviation less than 1 mm was considered highly reliable. The trueness of each scan application was deemed clinically acceptable for both diagnosis and treatment planning purposes [44]. A subgroup meta-analysis revealed a substantial difference in the accuracy of 3D facial scans performed on living subjects and inanimate items. This finding suggests that the results of research conducted in vitro or in a lab may differ from those acquired from human subjects [43]. According to the results of the present study, direct comparison with other studies is not feasible due to fundamental differences in study design. These differences include the use of either inanimate subjects or real patients, variations in the type of applications based on 3D depth-sensing cameras used, and differences in the number and location of surface areas evaluated.

While smartphone depth-sensing cameras operate on principles similar to those of professional laser scanning systems, it's worth noting that laser systems tend to be more sensitive to depths. This sensitivity arises from their construction with higher sensitivity sensors [22,25]. The literature has provided in vivo studies assessing the overall facial scanning accuracy of the Planmeca Proface laser scanner using reference anthropometric measurements. Menendez Lopez-Mateos et al. (2019) reported that the Planmeca scanner exhibited a mean error in accuracy of 1.04 mm when compared to direct anthropometric measurements. Similarly, Liberton et al. (2019) found that, except for a few landmarks around the eyes, the mean error in accuracy of this laser scanner, compared to two stereophotogrammetry systems, was less than 2.00 mm. In the present study, based on the facial partition divided into localized surface areas, both the nasal and perioral units exhibited significantly higher mean trueness values when calculated using the Planmeca ProFace scanner compared to the other scanners employed. Amornvit et al. (2019) [28] reported limitations of the Proface 3D Mid laser scanner, stating it was unable to scan undercuts wider than 6 mm at a depth exceeding 2 mm. Additionally, Koban et al. (2020) [41] noted lower accuracy in scanning nasal areas using a laser scanner. However, this specific finding was not corroborated in the present study. Based on the results of our study, we utilized a control mesh to compare against CBCT soft tissue segmentation obtained from the Planmeca Promax3D Mid cone beam scanner. This imaging unit, enhanced with the Planmeca ProFace scanner, has the capability to capture both a 3D photo and a CBCT image in a single rotation. This leads to improved trueness accuracy when assessing 3D deviation errors, particularly in the perioral region, where micro-motion movements or changes in resting lip position can be problematic. This improvement is achieved by utilizing face scans to test different machines at different time intervals. Aung et al. (1995) [33] also proposed that the orbital, circumoral, and nasal regions were reliable for laser scanners. Furthermore, Revilla-León et al. (2021) have revealed that

the position of the scanned surface area affected the accuracy of facial scanning for both trueness and precision. It was discovered that accuracy reduced when positioned more laterally and increased in the surface areas located closer to the facial midline.

Regarding the performance of the dual-structured light scanner in the present study, it was determined to have the least accuracy, with statistically significantly lower mean values for both trueness and precision compared to the EM3D Scanner application in terms of overall accuracy, encompassing both middle and lower facial regions. However, when investigating the scanning accuracy of the tested devices in separate facial regions, there were no statistically significant differences compared to the application in the surface areas located closer to the facial midline. The Einscan H2, a portable 3D scanner that uses a hybrid LED and infrared light source, has not yet been the subject of any research published in the literature that examines scanning accuracy. Its recent entry into the 3D face scanner market in 2023 is the reason for this. In order to compare with comparable technology, Piedra-Cascon et al. (2020) [27] utilized direct anthropometry as a reference method to evaluate the accuracy of a dual structured-light scanner connected to a tablet in generating 3D facial models of 10 individuals. Their mean accuracy of $0.910 \pm 0.320$ mm was deemed acceptable for virtual treatment planning, according to their findings. Liu et al. (2021) [45] demonstrated that both stationary stereophotogrammetry and a dual structured-light scanner system coupled to a tablet exhibited good accuracy and precision for clinical purposes when compared to direct anthropometry. Similar to this, in a recent paper by Cascos et al. (2023) [46], this technology achieved a mean accuracy value of 0.61 (±1.65) and 0.28 (±2.03) in maximum intercuspation and smile in sixty participants. However, Mai HN & Lee DH (2020) [43] suggest that when utilizing external structured-light scanners, the overall accuracy should be interpreted as a result that encompasses the performance of the compatible mobile or tablet device. In previous studies on handheld 3D structured light scanners from the Einscan lineup, Amornvit et al. (2019) [28] reported accuracy findings that differ from the results of our study. They found that the EinScan Pro 2X Plus and EinScan Pro structured light scanners exhibited significantly higher trueness values compared to the ProFace 3D Mid laser scanner and the iPhone X using the Bellus3D Face Application. Similarly, in the research conducted by Michelinakis et al. (2023) [47], the Einscan Pro HD scanner demonstrated a mean accuracy value of 0.358 mm, exhibiting significantly higher accuracy for the complete face and significantly higher trueness for each facial partition compared to other scanners such as the Planmeca Preface and Ray Face Scanner. One of the primary limitations of portable face-scanning systems is motion artifacts induced by involuntary facial movements and prolonged facial acquisition time. Due to the lack of clear control over scanning time in our study design, this factor may have been the main source of error in the results of these scanners.

The analysis of superimpositions revealed differences in reproducibility across various areas. Approximately 80% of the surface fell into the highly reproducible category, with the nose area scanned by the Planmeca ProFace scanner demonstrating the highest average percentage overlap value, followed by the chin and perioral area. For each of the three comparisons examined, between 80 and 90 percent of the five overlapping areas were within the tolerance limits of reproducibility. These outcomes align with what Pellitteri et al. (2021) [15] found. However, it appears that the percentages of locations within the tolerance range in our study are higher than theirs. The right cheek, at almost 60%, was the area that attained the highest percentage, followed by the chin and the tip of the nose. Pellitteri et al. (2023) [48] presented results that contrasted with previous findings. They demonstrated that the areas overlap analysis between scanners validated the accuracy of all systems, with over 90% of each area analyzed falling into the highly reproducible band. Additionally, the chin was found to be the most accurately recreated, with no variation observed between scanners.

The literature has reported the use of synthetic face markers to help with appropriate registration between the test and reference data sets [49,50]. On the patient's face, 4.00 mm diameter marker stickers were securely placed in various locations for use with the 3D scanners. Unfortunately, as illustrated in the diagrammatic representation of the proposed method. (Fig 1), the marker sticker on the 3D images obtained by the three different 3D scanners utilized in this study was unclear. It was clearly detectable only in the 3D image obtained from CBCT soft tissue segmentation. According to a previous study, while locating the spherical projections on the textured images could potentially offer a solution, they might

not consistently appear smaller and less visible than those on the non-textured 3D images. Additionally, their positions on the textured 3D images may not always align with the original balls [47,51]. However, the positions of the measurement points on the 3D image without texture did not always align with those on the textured image, especially when the distortion of the 3D image was significant due to the rendering process of attempting to overlay multiple photographs. This could result in pixel displacement while the surface volume of the image remained unchanged. As a result, the decision was made to refrain from using the marker stickers as facial indicators to aid in mesh registration. In our study, we applied the alignment algorithm and digitized surface area locations using the best-fit algorithm (BF). This method utilizes the iterative closest point (ICP) algorithm to align two meshes by minimizing the discrepancies between their point clouds. The algorithm continuously adjusts the transformation to reduce the error metric [52]. We specifically opted for the best-fit algorithm (BF) as processed directly by the software, without operator-defined alignment to specific sections of the dataset or pre-identified landmarks. This ensured a more standardized and automated approach to alignment, reducing potential human bias in the superimposition process. In the study by Revilla-Leon et al. (2021) [53], an in vitro experiment was conducted to evaluate the accuracy of a facial scanner based on different alignment methods, specifically the reference or section-based best fit (RBF) and landmark-based best fit alignment (LA) techniques. The RBF method aligns datasets by restricting the alignment to operator-identified sections of the dataset, while the LA method requires the operator to manually select common landmarks or points, which are then aligned by the software. Additionally, the study examined the digitized area, both total and localized, of a stereophotogrammetry scanner to assess its impact on alignment accuracy. The study suggested that when creating a virtual patient representation, where multiple scans need to be superimposed, the reference-based best fit (RBF) may be the most suitable alignment method. This recommendation is based on their findings that the RBF algorithm achieved higher trueness, but lower precision compared to the landmark-based best fit alignment technique. Additionally, they observed that the placement of the scanned surface area impacted the accuracy of facial scanning in terms of both trueness and precision. Specifically, it performed better in the center of the face but less accurately when positioned farther laterally. These findings align with our study results, which also demonstrated that the midface region consistently exhibited greater accuracy compared to the lateral areas across all three 3D face scanners used in our research.

Taking all factors into account, this study demonstrates that all three scanning systems used (laser scanner, dual-structured light, and smartphone) can be regarded as accurate means of obtaining 3D facial models. While statistically significant differences in accuracy among the three face scanners were detected, it remains uncertain whether these differences have clinical significance. All tested scanners had mean entire face trueness values that were within a 1 mm range. Deviations smaller than 1.0 mm were deemed extremely acceptable in clinical applications, particularly for activities such as facial aesthetic analysis in patient diagnosis and treatment planning. A facial model can be considered suitable in clinical practice if its variation is less than 1.5 mm [3,39]. Below this 1.5 mm clinical threshold, all the scanners used in the study exhibited deviations in complete face scanning trueness. The current study demonstrates that it is currently feasible to conduct 3D facial scanning in clinical settings using face scanners compatible with smartphones. The main advantages of these scanners are their portability and low cost. However, to minimize the risk of motion artifacts, patients must adhere more closely to instructions compared to other professional devices, requiring additional attention from the clinician. Patients need to remain motionless during the photo-acquisition procedure from various perspectives. However, in orthognathic, plastic, and maxillofacial rehabilitation cases, it is essential to reference soft tissue changes alongside changes in hard tissue and skeletal components post-treatment. Therefore, static face scanner devices capable of quickly capturing images, such as stereophotogrammetry scanners or CBCT imaging units incorporating integrated 3D face scan systems capable of capturing images in a single rotation, provide greater accuracy, particularly in surface areas closer to the facial midline, especially in the nose and perioral regions. This enhanced accuracy is due to their ability to minimize the risk of motion artifacts. There are limitations to the present study. Although the study included patients with skeletal

deformities undergoing orthognathic surgery, most of the sample consisted of patients with Class III skeletal deformities. This limitation prevents the assessment of differences in the accuracy of 3D images concerning facial deformities arising from other types of skeletal deformities. According to Zhao et al. (2017) [3], patients with facial deformities exhibit varying degrees of deflection and collapse. Specific areas such as the labiofacial sulcus, oral fissure, and angle of mouth experience an increase in undercut area due to the unusual shape of the facial tissue. This undercut region diminishes the precision of facial scanning and poses challenges for optical scanning. Future research should investigate the effectiveness of different facial scanning modalities in capturing various facial deformities and expressions, including those made during speaking or smiling. Ensuring accuracy when combining extraoral and intraoral data sets is crucial for this purpose. The development of virtual patient models can be facilitated by integrating tomographic data with static and dynamic surface data acquisition sets, which helps streamline the surgical treatment planning procedure.

## Conclusion

- The EM3D scanner application demonstrated the highest overall accuracy (trueness and precision) performance in both the middle face and lower face areas, yet there was no significant difference compared to the Planmeca ProFace scanner.

- Both the nasal and perioral units exhibited significantly greater trueness and precision values when calculated using the Planmeca ProFace scanner compared to the other scanners used, as determined by the facial partition separated into localized surface areas.

- An overall trueness ranging from 0.70 to 0.85 mm and an overall precision ranging from 0.68 to 0.81 mm were observed, with deviations less than 1.0 mm being considered highly acceptable. Additionally, clinical acceptable scanning accuracy was defined as less than or equal to 1.5 mm.

- The analysis of superimpositions uncovered variations in reproducibility across different areas. Around 80% of the surface fell into the highly reproducible category, with the nose area scanned by the Planmeca ProFace scanner exhibiting the highest average percentage overlap value, followed by the chin and perioral area.

## Supporting information

**S1 Table. Raw data used for the statistical analysis of the Intraclass Correlation Coefficient (ICC).** The data were obtained from ten randomly selected CBCT soft tissue segmentations. Each segmentation was manually marked for five localized surface areas, and the process was repeated twice by the same examiner, ensuring a minimum interval of seven days between each marking session.
(XLSX)

**S2 Table. Descriptive statistics of the trueness and precision values (mm) obtained from three-dimensional scanners across different areas.** T: Trueness, P: Precision, PMC: Planmeca Proface, Ein: EnScanH2, APP: EM3D scanner application, Rcheek: Right cheek unit, Lcheek: Left cheek unit, Nose: nose unit, Perioral: Perioral unit, Chin: Mental unit.
(PDF)

## Acknowledgments

The authors would like to acknowledge the Faculty of Dentistry, Mahidol University for providing support in the use of facilities and granting access to the Planmeca ProMax 3D ProFace for data collection from the study participants.

## Author contributions

**Conceptualization:** Nichakun Tangthaweesuk, Somchart Raocharernporn .

**Data curation:** Somchart Raocharernporn .

**Formal analysis:** Nichakun Tangthaweesuk.

**Investigation:** Nichakun Tangthaweesuk.

**Methodology:** Nichakun Tangthaweesuk.

**Supervision:** Somchart Raocharernporn .

**Writing – original draft:** Nichakun Tangthaweesuk.

**Writing – review & editing:** Nichakun Tangthaweesuk, Somchart Raocharernporn .

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
