## [Decision Letter · Decision Letter 0]

21 Jan 2025

PONE-D-24-47635THE ACCURACY OF THREE-DIMENSIONAL FACIAL SCAN OBTAINED FROM

THREE DIFFERENT 3D SCANNERS.PLOS ONE

Dear Dr. Raocharernporn ,

Thank you for submitting your manuscript to PLOS ONE. After careful consideration, we feel that it has merit but does not fully meet PLOS ONE’s publication criteria as it currently stands. Therefore, we invite you to submit a revised version of the manuscript that addresses the points raised during the review process.

We look forward to receiving your revised manuscript.

Kind regards,

Johari Yap Abdullah, B.S. & I.T, GradDip ICT, M.Sc, Ph.D.

Academic Editor

PLOS ONE

**Journal requirements:**

2. We note that Figures 1 and S1 includes an image of participant in the study. 

Reviewers' comments:

Reviewer's Responses to Questions

**Comments to the Author**

1. Is the manuscript technically sound, and do the data support the conclusions?

Reviewer #1: Partly

Reviewer #2: Yes

2. Has the statistical analysis been performed appropriately and rigorously? 

Reviewer #1: Yes

Reviewer #2: Yes

3. Have the authors made all data underlying the findings in their manuscript fully available?

Reviewer #1: No

Reviewer #2: Yes

4. Is the manuscript presented in an intelligible fashion and written in standard English?

Reviewer #1: No

Reviewer #2: Yes

5. Review Comments to the Author

**Reviewer #1: ** This study compared the accuracy, precision, and reproducibility of three 3D facial scanning systems: Planmeca Proface (laser scanner), EinScan H2 (structured light scanner), and EM3D Scanner (smartphone app). Thirty subjects with skeletal deformities were scanned, and results were compared to CBCT-derived facial surfaces. The EM3D Scanner outperformed the EinScan H2 in accuracy, particularly for the overall face, while the Planmeca Proface showed high accuracy in the nasal and perioral regions. Overall, most scans were reproducible, with deviations under 1 mm, making the EM3D Scanner and Planmeca Proface suitable for clinical use.

The article seems to be well presented, but some shortcomings can be identified.

Comments to the Author:

• Citing references in the text of the manuscript does not meet the journal's requirements.

• Then describing “experimental instruments”, you need to indicate what iPhone/iPad model you used with EM3D Scanner app directly in corresponding section

• There is no data provided about the scanned surfaces in STL format: number of edges, vertices. It is important then comparing scans of different scanners.

• I think, it is no reason to present RMS calculation formula if it was calculated by Geomagic X, not by you

• Intraexaminer reliability was assessed only with CBCT segmentation. 3D scanning quality is more dependent on the operator – why not to assess this iterexaminer reliability?

• Statistical analysis description could be more precise, clearer describing what analysis was used for what data (maybe separating by new sections).

• There are two same tables: Table 4 on pages 18 and 19.

• What are the meanings of superscripts a,b in Tables 4 to 6?

• Graphical representation of acquired/calculated data would be very useful – frequently it is more informative to the reader than just tables with numbers.

• Question about “marker stickers” – we can see on figures, what they are visible on reference 3D surfaces, but not on scanners surfaces (as you mention on page 30). From the images presented we can see that they are elevated or recessed and so they have impact on the data statistics. Could you comment on this?

• Language of the manuscript should be clearer, more concise and with no errors – moderate English language editing is needed.

• On page 13: “Two groups were generated, namely, the best-fit algorithm, which automatically determined the best-fit alignment by means of pre-established reference points.” – it is not clear about “two groups”.

• On page 14: “Furthermore, the following percentage of overlapping surfaces within the tolerance ranges was determined by the software: <…>” – then authors present the ranges of distances for reproducibility assessment, but from the sentence you expect “percentages”.

• “grater trueness” on page 24 should be changed to “grater trueness value”. Also, please clarify on page 29 “higher trueness compared”.

• “The PLOS Data policy requires authors to make all data underlying the findings described in their manuscript fully available without restriction <...>”. The full data used in the manuscript is not available directly – only “upon reasonable request”.

**Reviewer #2: ** The study compares the scanning accuracy of three different facial scanning devices using 30 subjects and holds clinical significance. However, there are several issues that require discussion with the authors:

1. Please supplement information on whether the scanning data were collected in a closed environment and whether the light source intensity was consistent during the scanning process for each device.

2. There is an error in the description of the results section: "the Einscan H2 obtained…0.85 ± 0.8 mm" should be corrected.

3. Table 4 appears duplicate in the manuscript.

4. The experimental results of this study indicate variations in scanning accuracy across different facial regions. How can the authors demonstrate that these differences in scanning accuracy are not caused by micro-movements of facial muscles?

5. Although the authors have mentioned various mesh data registration methods in the discussion, they have not clearly explained why the best-fit registration method was chosen. In fact, this registration method is not the most ideal, and the literature has mentioned registration methods based on the extraoral scan body (ESB). Please provide a comparison of these methods.

6. PLOS authors have the option to publish the peer review history of their article (what does this mean? ). If published, this will include your full peer review and any attached files.

**Do you want your identity to be public for this peer review?** For information about this choice, including consent withdrawal, please see our Privacy Policy .

Reviewer #1: No

Reviewer #2: No

---

## [Author Response · Author response to Decision Letter 1]

13 Feb 2025

We have made every effort to address all the comments and suggestions provided by both the Editor and Reviewers. These revisions are detailed in the attached file titled "Response to Reviewers," organized by each specific point raised.

If there are any remaining issues or misunderstandings regarding certain points that require further clarification or correction, please feel free to contact us, and we will promptly make the necessary adjustments.

We sincerely appreciate your consideration and acceptance of our research.

---

## [Decision Letter · Decision Letter 1]

27 Feb 2025

PONE-D-24-47635R1THE ACCURACY OF THREE-DIMENSIONAL FACIAL SCAN OBTAINED FROM THREE DIFFERENT 3D SCANNERS.PLOS ONE

Dear Dr. Raocharernporn ,

Thank you for submitting your manuscript to PLOS ONE. After careful consideration, we feel that it has merit but does not fully meet PLOS ONE’s publication criteria as it currently stands. Therefore, we invite you to submit a revised version of the manuscript that addresses the points raised during the review process.

We look forward to receiving your revised manuscript.

Kind regards,

Johari Yap Abdullah, B.S. & I.T, GradDip ICT, M.Sc, Ph.D.

Academic Editor

PLOS ONE

Journal Requirements:

Reviewers' comments:

Reviewer's Responses to Questions

**Comments to the Author**

1. If the authors have adequately addressed your comments raised in a previous round of review and you feel that this manuscript is now acceptable for publication, you may indicate that here to bypass the “Comments to the Author” section, enter your conflict of interest statement in the “Confidential to Editor” section, and submit your "Accept" recommendation.

Reviewer #1: All comments have been addressed

Reviewer #2: All comments have been addressed

2. Is the manuscript technically sound, and do the data support the conclusions?

Reviewer #1: (No Response)

Reviewer #2: Yes

3. Has the statistical analysis been performed appropriately and rigorously? 

Reviewer #1: (No Response)

Reviewer #2: Yes

4. Have the authors made all data underlying the findings in their manuscript fully available?

Reviewer #1: (No Response)

Reviewer #2: Yes

5. Is the manuscript presented in an intelligible fashion and written in standard English?

Reviewer #1: No

Reviewer #2: Yes

6. Review Comments to the Author

Reviewer #1: I think the English still needs editing.

Also check formatting then you are citing publications, as example: "sensitivity sensors. [22,25]" -> sensitivity sensors [22,25]." - see examples of previously published articles.

Reviewer #2: (No Response)

7. PLOS authors have the option to publish the peer review history of their article (what does this mean? ). If published, this will include your full peer review and any attached files.

**Do you want your identity to be public for this peer review?** For information about this choice, including consent withdrawal, please see our Privacy Policy .

Reviewer #1: No

Reviewer #2: No

---

## [Author Response · Author response to Decision Letter 2]

4 Mar 2025

We extend our sincere gratitude to Reviewer #1 for dedicating their time to thoroughly reviewing our research and offering insightful suggestions to enhance the quality of the manuscript.

We have thoroughly reviewed and corrected the grammar and language in the manuscript to ensure clarity and accuracy. Additionally, we have revised and improved the formatting of cited publications in accordance with your recommendations.

---

## [Decision Letter · Decision Letter 2]

21 Mar 2025

THE ACCURACY OF THREE-DIMENSIONAL FACIAL SCAN OBTAINED FROM

THREE DIFFERENT 3D SCANNERS.

PONE-D-24-47635R2

Dear Dr. Raocharernporn ,

We’re pleased to inform you that your manuscript has been judged scientifically suitable for publication and will be formally accepted for publication once it meets all outstanding technical requirements.

Kind regards,

Johari Yap Abdullah, B.S. & I.T, GradDip ICT, M.Sc, Ph.D.

Academic Editor

PLOS ONE

Additional Editor Comments (optional):

Reviewers' comments:

Reviewer's Responses to Questions

**Comments to the Author**

1. If the authors have adequately addressed your comments raised in a previous round of review and you feel that this manuscript is now acceptable for publication, you may indicate that here to bypass the “Comments to the Author” section, enter your conflict of interest statement in the “Confidential to Editor” section, and submit your "Accept" recommendation.

Reviewer #1: All comments have been addressed

2. Is the manuscript technically sound, and do the data support the conclusions?

Reviewer #1: Yes

3. Has the statistical analysis been performed appropriately and rigorously? 

Reviewer #1: Yes

4. Have the authors made all data underlying the findings in their manuscript fully available?

Reviewer #1: Yes

5. Is the manuscript presented in an intelligible fashion and written in standard English?

Reviewer #1: Yes

6. Review Comments to the Author

Reviewer #1: (No Response)

7. PLOS authors have the option to publish the peer review history of their article (what does this mean? ). If published, this will include your full peer review and any attached files.

**Do you want your identity to be public for this peer review?** For information about this choice, including consent withdrawal, please see our Privacy Policy .

Reviewer #1: No

---

## [Editor Report · Acceptance letter]

PONE-D-24-47635R2

PLOS ONE

Dear Dr. Raocharernporn ,

I'm pleased to inform you that your manuscript has been deemed suitable for publication in PLOS ONE. Congratulations! Your manuscript is now being handed over to our production team.

Kind regards,

on behalf of

Dr. Johari Yap Abdullah

Academic Editor

PLOS ONE